# Effects of Molecular Iodine/Chemotherapy in the Immune Component of Breast Cancer Tumoral Microenvironment

**DOI:** 10.3390/biom11101501

**Published:** 2021-10-12

**Authors:** Olga Cuenca-Micó, Evangelina Delgado-González, Brenda Anguiano, Felipe Vaca-Paniagua, Alejandra Medina-Rivera, Mauricio Rodríguez-Dorantes, Carmen Aceves

**Affiliations:** 1Instituto de Neurobiología, Universidad Nacional Autónoma de México, Querétaro 76230, Mexico; olgacuenca76@gmail.com (O.C.-M.); edelgado@comunidad.unam.mx (E.D.-G.); anguianoo@unam.mx (B.A.); 2Unidad de Biomedicina, Facultad de Estudios Superiores Iztacala, Universidad Nacional Autónoma de México, Tlalnepantla 54090, Mexico; felipe.vaca@iztacala.unam.mx; 3Laboratorio Nacional en Salud, Diagnóstico Molecular y Efecto Ambiental en Enfermedades Crónico Degenerativas, Facultad de Estudios Superiores Iztacala, Universidad Nacional Autónoma de México, Tlalnepantla 54090, Mexico; 4Subdirección de Investigación Básica, Instituto Nacional de Cancerología, Mexico City 14160, Mexico; 5Laboratorio Internacional de Investigación sobre el Genoma Humano, UNAM-Juriquilla, Querétaro 76230, Mexico; amedina@liigh.unam.mx; 6Instituto Nacional de Medicina Genómica, Mexico City 14610, Mexico; mrodriguez@inmegen.gob.mx

**Keywords:** molecular iodine, immune response, breast cancer

## Abstract

Molecular iodine (I_2_) induces apoptotic, antiangiogenic, and antiproliferative effects in breast cancer cells. Little is known about its effects on the tumor immune microenvironment. We studied the effect of oral (5 mg/day) I_2_ supplementation alone (I_2_) or together with conventional chemotherapy (Cht+I_2_) on the immune component of breast cancer tumors from a previously published pilot study conducted in Mexico. RNA-seq, I_2_ and Cht+I_2_ samples showed significant increases in the expression of Th1 and Th17 pathways. Tumor immune composition determined by deconvolution analysis revealed significant increases in M0 macrophages and B lymphocytes in both I_2_ groups. Real-time RT-PCR showed that I_2_ tumors overexpress T-BET (*p* = 0.019) and interferon-gamma (IFNγ; *p* = 0.020) and silence tumor growth factor-beta (TGFβ; *p* = 0.049), whereas in Cht+I_2_ tumors, GATA3 is silenced (*p* = 0.014). Preliminary methylation analysis shows that I_2_ activates IFNγ gene promoter (by increasing its unmethylated form) and silences TGFβ in Cht+I_2_. In conclusion, our data showed that I_2_ supplements induce the activation of the immune response and that when combined with Cht, the Th1 pathways are stimulated. The molecular mechanisms involved in these responses are being analyzed, but preliminary data suggest that methylation/demethylation mechanisms could also participate.

## 1. Introduction

The immune component of the tumor microenvironment is considered one of the key players in prognosis and response to treatment [1]. In recent decades, the intratumoral presence of immune cell phenotypes has been associated with the prognosis of the disease. Thus, cytotoxic lymphocytes (Th1 and CD8+), M1 macrophages, and their effector molecules are considered favorable prognostic indicators [2], while immunomodulatory lymphocytes (Th2, Treg) and macrophages (TAM-M2) are found in worse prognosis scenarios [3]. At the molecular level, cytokines IL-1, IL-6, and tumor growth factor-beta (TGFβ) are associated with tumor progression, whereas IL-12 and interferon-gamma (IFNγ) can inhibit cancer proliferation and/or metastasis [4]. Immune cells can switch these secretion patterns from one lineage towards another under certain circumstances, exhibiting phenotypic plasticity [5]. This functional switch, or trans-differentiation, depends on epigenetic processes [6]. Methylation/demethylation of DNA is an epigenetic mechanism concerning the transfer or removal of a methyl group onto the C5 position of the cytosine. Methylation regulates gene expression by recruiting proteins associated with gene repression or inhibiting the binding of transcription factors to DNA [7]. Conversely, active demethylation allows gene activation [6]. In the antitumor immune response, demethylation of the IFNγ locus activates the transition from naïve to memory CD8+ T cells, promoting increased IFNγ secretion [8]. Some dietary compounds can modify cancer progression, and over the past decade, numerous micronutrients have demonstrated activity as epigenetic modulators [9]. Molecular iodine (I_2_) exerts antineoplastic effects on different cancer models [10,11], whereas in its non-oxidized form, like iodide (I-) or thyroid hormones (T4), it is not able to achieve these effects [12]. In cancer cells, I_2_ could act as a “mitocan” agent (acronym for mitochondria and cancer) by depleting thiol reserves or disturbing the mitochondrial membrane potential (Mmp), thereby inducing apoptotic pathways [13]. Additionally, this chemical form of iodine is an effective antioxidant, even tenfold more effective than ascorbic acid [14]. Moreover, I_2_ exhibited indirect antitumor activity by generating 6-iodolactone (6-IL) through the iodination of arachidonic acid. This iodolipid is an active ligand of peroxisomal-activated receptor type gamma (PPARγ), inducing re-differentiation by inhibiting stem signaling and triggering apoptosis [15]. In addition, I_2_ supplementation exerts effects on the immune system, acting as a direct genetic modifier [16] or as an attractor, increasing the amount of CD8+ lymphocytes within the tumor [17]. We previously demonstrated in a breast cancer pilot study that I_2_ supplementation exerted adjuvant effects when combined with conventional chemotherapy, reducing the residual tumor size, and increasing disease-free survival [17]. The RNA-seq analysis showed that I_2_-treated tumors exhibited significant activation of Th1, NK, and CD8 cytotoxicity pathways [17]. In the present study and using the same transcriptomic bank, we analyzed I_2_ and the chemotherapy treatment (Cht) in the immune scenario. We describe the epigenetic patterns of immune effectors at the methylation and demethylation level.

## 2. Materials and Methods

### 2.1. Mammary Tumors

Tumors were collected as part of a pilot study registered at Clinicaltrial.gov (NCT03688958). Briefly, two pilot study groups were established based on the stage of cancer diagnosed: Early (stage II) and Advanced (stage III) breast cancer groups. Thirty patients were randomly assigned (double-blind) to receive either molecular iodine (I_2_; 5 mg/day) or a placebo (vegetable colored water) for 7–35 days (as determined by the preoperative oncologist’s protocol). In the Advanced group, 30 patients were randomly (double-blind) divided into the I_2_ or placebo groups, and both groups received 4–6 cycles of neoadjuvant chemotherapy (Cht; 5-fluorouracil/epirubicin/cyclophosphamide or taxotere/epirubicin). Daily, after breakfast, I_2_ or placebo was diluted in drinking water. During the surgical procedure, the tumor sample was kept in dry ice to avoid degradation and stored at −80 °C until further analysis.

### 2.2. RNA-Seq and Transcriptomic Analysis

Detailed constructions and all specific data analyses, including pathway and upstream regulator prediction, as well as all other analyses involving the transcriptomic data, can be found in protocols.io [18]. Briefly, total RNA was extracted with Qiazol and RNeasy (both from Qiagen, Valencia, CA, USA). Two different pools of four individual tumor samples were used. As a normal control, we used a pool of two normal mammary gland samples from aesthetic surgeries (volume reduction). Poly-A enriched mRNA was used to construct stranded mRNA-Seq libraries following the manufacturer’s instructions (KAPA Biosystems). Sequencing was carried out at Duke University Genome Sequencing Shared Resource Center (Durham, NC, USA). The libraries were sequenced on an Illumina HiSeq 2500 platform, in which 101 bases were determined in pair-end mode. Data were assessed for quality and trimmed with FastQC (Version number 0.11.7, Cambridge, UK) and Trimmomatic (Version number V0.32, Mühlenberg, Germany), respectively. Reads were mapped to the human genome (GRCh38), and expression levels were determined by htseq-count. Differential expression analysis was performed using Fisher’s exact and Benjamini–Hochberg (FDR) tests. Genes that were altered at least 2-fold or less than 0.5-fold with an FDR value equal to or lower than 0.05 were considered biological and statistically significant. The complete annotated sequences from the RNA-sequencing are available at the European Nucleotides Archives website (https://www.ebi.ac.uk/ena/erp110028) (accessed on August 2019).

### 2.3. Gene Set Enrichment Analysis

Gene set enrichment analysis (GSEA) was performed with Webgestalt (2013, Houston, TX, USA) and GSEA with the following parameters: Organism of Interest: hsapiens; Method of Interest: GSEA; Functional Database: pathway, Kegg; Select Gene ID Type: genesymbol. Annotation of genes with immunological function was done with the Gene Ontology Consortium (wiki.geneontology.org/index.php/Immunology) (accessed on June 2018). The 1325 most relevant immune genes were selected.

### 2.4. Th1 and Th2 Differentiation Genes

Genes known to be involved in CD4+ T cell differentiation towards Th1 or Th2 cells were obtained from public datasets (KEGG hsa04658, R&D systems Pathways) and were analyzed in our differential expression gene sets.

### 2.5. Deconvolution Analysis

Deconvolution studies were performed with CIBERSORT [19], which accurately quantifies the relative levels of different types of immune cells within a complex mixture of gene expression, and we used GED-IT to predict the cell type composition of tissue samples. We also used ICTD to deconvolute and identify immune cells [20].

### 2.6. Real Time RT-PCR

Gene expression was quantified with the real-time quantitative polymerase chain reaction (qPCR) method previously described [21]. Total RNA was obtained according to the protocol described by the manufacturer (TRIzol reagent, Life Technologies, Inc., Carlsbad, CA, USA). Messenger RNA (2 mg) was reverse transcribed using oligo-deoxythymidine primers. Each PCR was done using a specific pair of oligonucleotides detailed in Appendix A. A Rotor-Gene 3000 apparatus (Corbett Research, Mortlake, NSW, Australia) was employed to perform qPCR with a marker for DNA amplification (SYBR Green, Fermentas, Burlington, ON, Canada). Gene expression was calculated by the 2-DDCT method and was normalized to the housekeeping gene β-actin. Appendix A shows the oligos used for these amplifications.

### 2.7. Immunohistochemistry

The tumor tissues were cut into sections of 4 µm and treated with 3-aminopropyl-triethoxysilane for subsequent staining with hematoxylin and DBA or with specific antibodies for T-BET and IFNγ. Quantification of lymphocytes or positive-stain cells was performed using ImageJ software (Version 1.41, NIH, Bethesda, MD, USA) from three different sections of each tumor at 40× and 63×. We analyzed three tumors per experimental group.

### 2.8. Methylation-Specific PCR

Tumor DNA extraction and purification (Quick-DNA Miniprep Plus Kit, Zymo, CA, USA and DNA Clean & Concentrator-25, Zymo, CA, USA) was performed following the manufacturer’s instructions. Subsequently, the DNA was subjected to sodium bisulfite transformation (EZ-96 DNA Methylation MagPrep, Zimo, CA, USA). Promoter regions with CpG islands (FASTA and Methprime) were identified, and differential oligos for IFNγ and TGFβ were generated for these M&U regions (Appendix A). Amplification was performed with both oligos together with the housekeeping gene MLH-1 in endpoint PCR. Subsequently, a nested q-PCR was performed using 4 µL of the product of the first amplification. Relative gene expression was calculated by the 2-DDCT method and was normalized to the housekeeping gene MLH-1.

### 2.9. Statistic Analysis

Differential expression analysis for RNA-Seq was performed using Fisher’s exact and Benjamini–Hochberg (FDR) tests. Genes that were altered at least 2-fold or less than 0.5-fold with an FDR value equal to or lower than 0.05 were considered biological and statistically significant. Individual gene expressions and methylated/unmethylated amplicon amounts were analyzed with Student’s *t*-test between treatment and control samples. *p*-values less than 0.05 were considered statistically significant.

## 3. Results

### 3.1. Supplementation with I_2_ Increases the Immune Pathways Associated with an Antitumor Response

We first evaluated the expression level of genes involved in the immune response in the early and advanced tumors as compared with the normal tissue controls. As shown in Figure 1, regardless of tumor stage, I_2_ supplementation activates Th1 and Th17 antitumor differentiation pathways, T receptor cells, NK cytotoxicity, B cell receptor, and antigen processing/presentation. The color kay showed a genetic overexpression at least two times higher in advanced tumors (Ch+I_2_) than in early-stage tumors (I_2_).

### 3.2. I_2_ Increases the Intratumoral Ratio of Antigen-Presenting Macrophages/Dendritic Cells in Early-Stage Tumors and B Lymphocytes in the Advanced Stage

To identify the immune cell composition of the infiltrate, we conducted a deconvolution analysis with the CIBERSORT and ICTD algorithms. The analysis was carried out on two pools (5 or 6 different samples in each pool) for each experimental group and very similar results were observed in each pool-group. Both software programs showed an increased relative percentage of macrophages-dendritic cells in early tumor samples (placebo and I_2_). In advanced tumor samples (Cht and Cht+I_2_), both software programs concur with an increasing of CD4 T and B cells global relative percentage (Figure 2A,B). When analyzing the effect of the I_2_ supplement on the proportions of intratumoral immune cells, different results were found depending on the software applied. The CIBERSORT analysis (Figure 2A) showed an increase in the relative number of macrophages M0, while ICTD interpreted this increase as dendritic cells in early stages (Figure 2B). In the case of advanced-stage tumors, supplementation with I_2_ increased the fraction of B cells, pointing to an activation of the tumoral response in the presence of both components (Cht+I_2_).

### 3.3. I_2_ Activates Th1 Differentiation in the Early Stages of the Disease, While in Advanced Stages It Suppresses Th2 Differentiation

To corroborate the results obtained with the RNA-seq experiments, individual tumor samples were used to analyze markers of the cytotoxic IL12RB1, T-BET, IFNγ, and oncogenic GATA3 and TGFβ inducers. Figure 3 shows that I_2_ supplementation is accompanied by a significant increase in the expression of T-BET and IFNγ, and by the repression of TGFβ in early-stage tumors (I_2_). In advanced-stage tumors, I_2_ generates a decrease in the Th2 polarization marker GATA3 (Cht+I_2_). These data indicate that the presence of iodine at any stage induces an oncogenic polarization through Th1. The overexpression of T-BET and IFNγ was also detected at the protein level in tumor tissues of early-state patients supplemented with I_2_ compared to placebo (Figure 4).

### 3.4. I_2_ Modifies the Epigenetic Landscape Activating Antitumor Gene Promoters Expression and Silencing Oncogenic Genes

To further investigate the molecular mechanisms of I_2_ in the tumors, we evaluated the methylation status of IFNγ and TGFβ gene promoters with specific primers for the unmethylated (active) and methylated (inactive) states. Figure 5A shows that in early-stage tumors (placebo and I_2_), there were no significant differences between unmethylated or methylated forms. In contrast, in the advanced-stage tumors, the presence of chemotherapy is accompanied by the absence of active IFNγ (unmethylated) and a significant number of active forms of TGFβ (unmethylated). In these conditions, the presence of I_2_ (Cht+I_2_) showed changes through the highest levels of active IFNγ (*p* > 0.051) and a total suppression of TGFβ (undetectable amount of unmethylated form; <0.049). These patterns become more evident when we analyze the unmethylated/methylated index (division between the mean of unmethylated/methylated amplicons of each group; Figure 5B), showing that in tumors that were supplemented with both components (Cht+I_2_), I_2_ redirected the activation of the Th1 antitumor pathway through epigenetic mechanisms.

## 4. Discussion

Avoiding immune destruction and tumor-promoting inflammation in the tumor microenvironment are hallmarks of cancer initiation, and the immune component plays a key role in progression and metastasis [22]. Recognition of the critical importance of the microenvironmental component has resulted in a shift in therapeutic strategies, placing greater emphasis on treatments that include its modulation. While CAR-T cells and CTLA-4 and PD-1 blocking therapies are currently the most effective ways to reactivate the antitumor immune system, other components, some of natural origin, can reactivate the antitumor immune system and improve conventional therapies [23].

Molecular iodine is a micronutrient that shows antineoplastic properties in preclinical and clinical studies of breast cancer [10,24,25]. The mechanisms of action include direct antioxidant actions such as scavenging ROS and modulating mitochondrial functionality, as well as indirect actions activating PPARγ receptors, triggering apoptosis, and cell redifferentiation [13,17,26,27,28]. In a previous analysis of this protocol, it was demonstrated that the I_2_ supplement plus chemotherapy generated the best antitumor response (smaller tumor size and cancellation of chemoresistance) and increased the disease-free survival from 63 to 92% in five years in patients who received the I_2_ supplement before and after surgery [17]. Transcriptomic analysis showed that I_2_ promoted the antitumor response (Th1), increasing the presence and cytotoxic activity of intratumoral NK and CD8 + cells. In the present work, the specific analysis of the immunological profile showed that I_2_ generally activates both the anti-oncogenic and oncogenic immune pathways (Th1, Th17), and that the presence of chemotherapy enhances the antitumor effect of I_2_, as the response scale in these tumors (Cht+I_2_) was more than double.

Deconvolution analysis showed that I_2_ increases the amount of M0 (or dendritic cells) and B lymphocytes, corroborating the preponderance of the antitumor response. The two subtypes of augmented B cells were naïve B cells and memory B cells. Activated naïve B cells have been shown to promote Th1 polarization [29], while memory B cells can mount a rapid antibody response, effectively controlling tumor growth [30]. Lymphocytes and macrophages are highly plastic cells that can change their phenotype in response to their microenvironment [31,32]. Increased IFNγ synthesis has been associated with a better prognosis both by the inhibition of Th2 and M2 oncogenic immune polarization [33] and by decreased angiogenic capacity [34]. Our results not only show an increase in the mRNA expression and protein content of IFNγ, but also the upstream activation of the Th1 pathway via expression of T-BET, which is the main regulator of IFNγ. T-BET (encoded by *TBX21*) is an immune cell-specific member of the T-box family of transcription factors. It is expressed in a variety of immune cells, including dendritic cells, NK, CD4+, and CD8+, B cells, and a subtype of Tregs. T-BET+ cells function as antitumor lymphocytes by enhancing the production of cytokines such as IFNγ [35]. Previous studies have shown that the presence of intratumoral T-BET+ lymphoid cells correlate with a good prognosis in all breast cancers [36]. We discovered that the Cht+I_2_ combination not only promotes Th1 expression patterns in advanced-stage tumors, but also induces the silencing of key Th2 players such as GATA3. This transcription factor plays a critical role in the development of T cells in the thymus. Moreover, GATA3 controls the differentiation of naïve CD4 T cells and induces remodeling of the chromatin loci of Th2 cytokines and is an active repressor of IFNγ expression [37]. The mechanisms by which I_2_ induces this transdifferentiation effect in the tumor microenvironment have received scant attention. However, it is well described that immune modulation components are regulated by epigenetic mechanisms, where natural factors derived from the diet could take part [6,38]. In fact, in cancer progression, many of the changes in expression patterns are regulated at the epigenetic level by methylation/demethylation in gene promoters [39]. Recently, ascorbic acid has received great attention since this micronutrient participates as a cofactor of TET enzymes (ten eleven translocations) involved in histone and DNA demethylation and, therefore, in the epigenetic regulation of gene expression [40]. TET proteins convert 5-methylcytosine (5mC) to 5-hydroxy-methylcytosine (5hmC), 5-formylcytosine (5fC), and finally to 5-carboxytosine (5caC). Then, 5fC and 5caC are replaced by cytosine by base cleavage repair machinery [41]. Ascorbic acid increases TET-dependent 5hmC production and induces cytosine demethylation in mammals [42]. Furthermore, in a lymphoma mouse model, the intratumoral epigenome revealed a global increase of 5hmC after ascorbic acid treatment in the presence of PD1, suggesting a direct effect of ascorbic acid on CD8+ T cells and their cytotoxic function [43]. Interestingly, I_2_ exerts antioxidant effects in the same way as ascorbic acid does, by producing electrons, and in ferric reactions that measure its capacity, I_2_ is 10 times more potent than ascorbic acid [14]. To the best of our knowledge there are currently no studies examining the role of I_2_ in the functionality of TETs. In conclusion, the preliminary findings from this study indicate that I_2_, when used in conjunction with conventional chemotherapy, induces immune activation and redirects the response to the Th1 pathway through methylation and demethylation mechanisms.

## Figures and Tables

**Figure 1 biomolecules-11-01501-f001:**
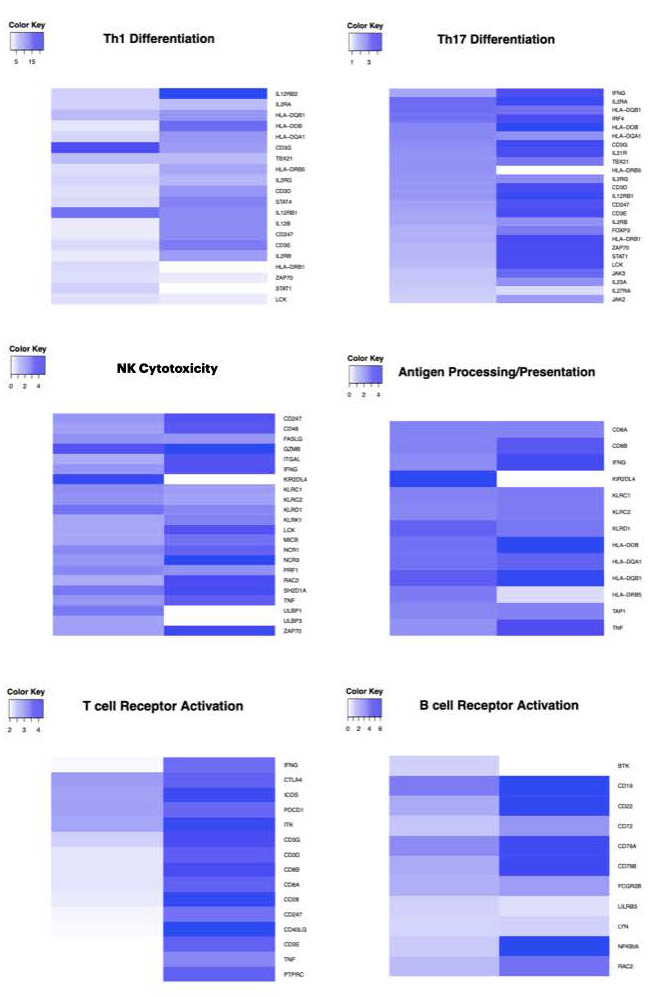
Immune pathways activated by iodine supplementation. The expression of genes in the I_2_ group correspond to early-stage tumors and those of the Cht+I_2_ group correspond to the advanced-stage tumors.A color scale (color key) specific for each pathway is depicted. The overexpressed genes for each pathway are shown in the right axis of each heatmap.

**Figure 2 biomolecules-11-01501-f002:**
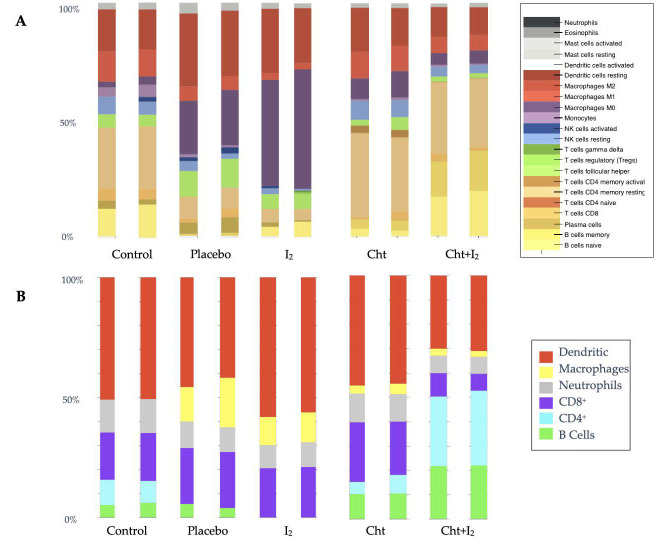
Deconvolution analysis of the relative composition of immune cells from two different pools of samples of each group. (**A**) Deconvolution performed with CIBERSORT. (**B**) Deconvolution obtained with ICTD. In both panels, Control bar corresponds to the non-cancer tissue (normal breast sample pool), placebo and I_2_ correspond to early-stage tumors and Cht and Cht+I_2_ to those in advanced stages. The composition of the immune infiltrate is color-coded and presented on the right side of each panel.

**Figure 3 biomolecules-11-01501-f003:**
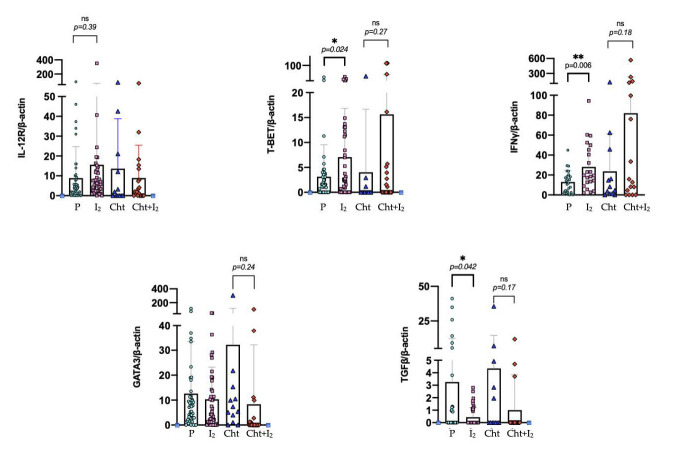
Gene expression of tumor suppressor cytotoxic and oncogenic inducers in individual samples. Expression was measured at the mRNA level by RT-qPCR. Data represent mean ± SD of three independent experiments from three individual samples. Significant values correspond to a Student’s t-test between I_2_ and its respective control group (* *p* < 0.05, ** *p* < 0.01). P: Placebo; I_2_: Iodine; Cht: Chemotherapy and Cht+I_2_: Chemotherapy and Iodine.

**Figure 4 biomolecules-11-01501-f004:**
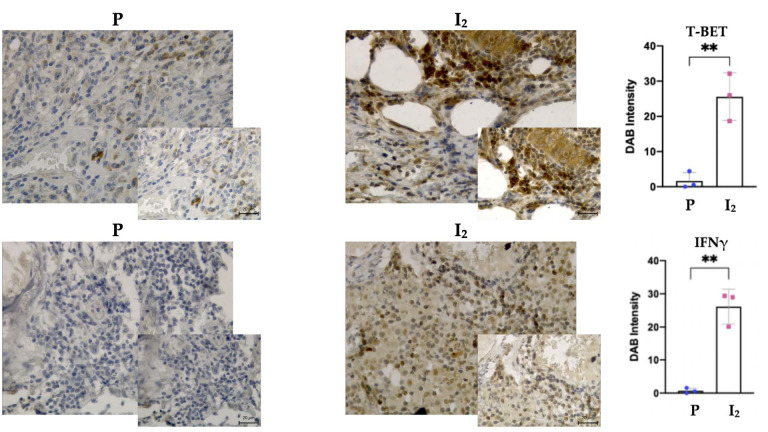
Protein expression of T-BET and IFNγ in individual samples of tumor tissue from early-stage patients (P; Placebo, I_2_; iodine). Data represent mean ± SD of three independent immunochemistry experiments from three individual samples. Student’s *t*-test ** *p* < 0.05.

**Figure 5 biomolecules-11-01501-f005:**
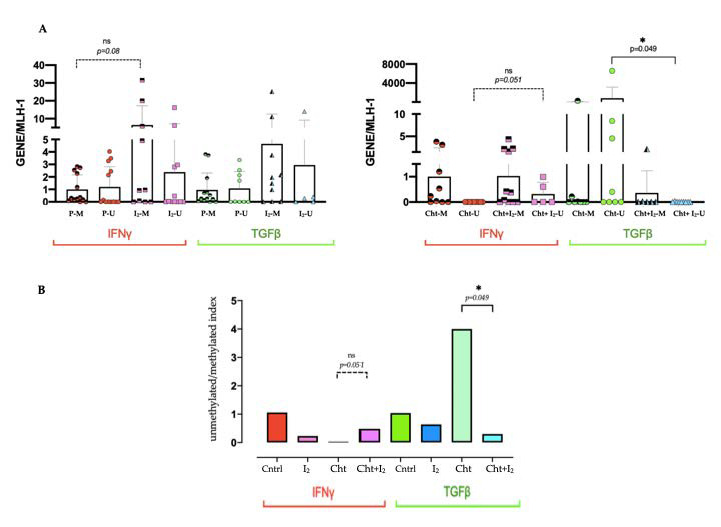
Methylation pattern of IFNγ and TGFβ gene promoters. (**A**) Amplification of the promoters (qPCR) of Unmethylated (U) or Methylated (M) forms in individual samples. The quantification was normalized by the expression of the housekeeping gene MLH-1. Left panel stage II and right panel stage III. (**B**) Unmethylated/Methylated index (means division) of each gene. Cntrl, control; I_2,_ iodine; Cht, chemotherapy; Cht+I_2_, Chemotherapy plus iodine. Student’s *t*-test * *p* < 0.05.

## Data Availability

The complete annotated sequences from the RNA-sequencing are available at the European Nucleotides Archives website (https://www.ebi.ac.uk/ena/erp110028 accessed on 8 September 2021).

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
