# Peer review of "Effects of Molecular Iodine/Chemotherapy in the Immune Component of Breast Cancer Tumoral Microenvironment"

_biomolecules, 2021, doi:10.3390/biom11101501_

Round 1

Reviewer 1 Report

A very interesting and well done study concerning the ability of the I2 to act as drug supplements in biomedical applications is reported. All conclusions well fit with the obtained experimental results. In my opinion the manuscript should be published in Biomolecules. 1. Figure 1 should be clean given, however, unclear text is showed. 2. In materials and methods part, the Cht represent 5-fluorouracil/epirubicin/cyclophosphamide or taxotere/epirubicin, the ratio and amount of fluorouracil/epirubicin/cyclophosphamide or taxotere/epirubicin should be added. 3. In Figure 2 and 3, Cht+I2 is 5-fluorouracil/epirubicin/cyclophosphamide or taxotere/epirubicin and the detail should be added. Meanwhile, the results of deconvolution analysis and gene expression for 5-fluorouracil/epirubicin/cyclophosphamide+I2 group and taxotere/epirubicin+I2 group should be showed. 4. Why choose oral administration and what is the difference to injection administration? 5. How to determine the dose 5 mg/day?

Author Response

A very interesting and well-done study concerning the ability of the I2 to act as drug supplements in biomedical applications is reported. All conclusions well fit with the obtained experimental results. In my opinion the manuscript should be published in Biomolecules.

  1. Figure 1 should be clean given; however, unclear text is showed.

We thank the referee for the observation, in the new version we include the description of the figure.

2. In materials and methods part, the Cht represent 5-fluorouracil/epirubicin/cyclophosphamide or taxotere/epirubicin, the ratio and amount of fluorouracil/epirubicin/cyclophosphamide or taxotere/epirubicin should be added.

In the previous publication, we have a table (supplementary table 3) that summarizes the initial treatments of these patients and only 4 of the 30 patients received TE (3 with iodine and 1 with placebo), the rest received FEC (between 4 to 6 cycles) for therefore we do not have enough numbers to do a differential analysis. Furthermore, in all cases these patients received a second round of chemotherapy + iodine after mastectomy. In this second round, 40% (15 patients) received TE. Our analysis of tumor response, survival, genes, etc., was not associated with the specific treatment. For that reason, in the present work we described as a single treatment (chemotherapy; Cht).

  1. In Figure 2 and 3, Cht+I2 is 5-fluorouracil/epirubicin/cyclophosphamide or taxotere/epirubicin and the detail should be added. Meanwhile, the results of deconvolution analysis and gene expression for 5-fluorouracil/epirubicin/cyclophosphamide+I2 group and taxotere/epirubicin+I2 group should be showed.

See the answer 2.

  1. Why choose oral administration and what is the difference to injection administration?
  2. How to determine the dose 5 mg/day?

Iodine in its molecular form (I2) or as a Lugol solution (I2 + KI mixture) has been extensively studied in preclinical (rodents and canines) and clinical (mastalgia, prostatic hyperplasia, and breast cancer) studies. It is well known that 3 to 6 mg/day is the dose that exerts antioxidant, anti-inflammatory, and antiproliferative effects. They do not generate any side effects on thyroid function or health in general. Molecular iodine or Lugol are ingested orally diluted in water. Molecular iodine is very reactive and causes severe skin or injection site irritation. To review these effects and doses, you can consult the following citations.

Ghent, et al. Can. J. Surg. 36, 453–460 (1993)

Kessler, J.H. Breast J., 10, 328–336 (2004)

García-Solís, et al. Mol. Cell. Endocrinol. 236, 49–57 (2005).

Aceves, et al. Thyroid 23, 938–946 (2013).

Zambrano et al.  BMC Vet. Res. 14, 87 (2018).

Quintero et al. Free Radical Biol Med 115, 298–308 (2018)

Moreno-Vega, A. et al. Nutrients 11, 1623-1642 (2019).

Reviewer 2 Report

The present manuscript is interesting and describe an interesting methodology to determine the immune population present in cancer tissue as well as the potential antituor role of molecular iodine.

Introduction and discussion are well presented and structured, the subject flow and the background and future directions are well discussed.

However, Material and Methods and Results section must be improved.

Matherial and Methods:

  • In the Real Time RT-PCR it is not clear how the authors calculated the relative gene expression. If a comparative DDCt method was used then normalization with b-actin is used; otherwise if a standard curve was used, than an absolute quantification was done. Please clarify and correct accordingly.
  • A Statistical analysis paragraph should be added 

Results:

  •  Reference to Figure 1 is missing in the text
  • Figure 1 is difficult to understand. First of all image qualitymust be improved and fonts of the figure must be enlarged as it is impossible to read the gene names. It is not clear what the two columns of each heatmap refer to, a lable over the graph shuld be added. Figure 1 legend is not self explanatory and some information are missing. Color code of the heatmap are nt clear: does withe means that the gene is not overexpressed and thus not regulated in tumor? Then how can the authors state that "Genetic overexpression was found to be twofold higher in advanced tumors than in early-stage tumors"? Additionally, as said, it is not clear what refers to early stage tumor, what to advanced tumor, what to I2 treatment and what to Cht+I2: this is very confusing.
  • Sentence: "I2 increases the intratumoral ratio of antigen-presenting macrophages/dendritic cells...": nowhere a ratio between antigen-presenting macrophages/dendritic cells has been measured or expressed. Please remove the word ratio
  • Sentence: "With both methods, we observed that the treatments (I2 and Cht) are accompanied by an increase in the levels of M2 macrophages...." It is not correct to say with both methods, as for what  I could understand, the ICTD methods cannot discriminate between macrophages subpopulations
  • Also figure 2 is difficult to understand and it is not clear what the columns of the two panels refer to: What the two columns for each treatment refer to? Please indicate it more clearly in the graph. Where are the reference to early stage and late stage groups? it is not clear. It was very hard to realte what said in the text with what was in the figure. Please try to facilitate the reading by making the figure easier to understand and the text more explanatory.
  • Figure 4: please inidcate which protein is analysed in the first line of IHC and which in the second line of IHC. The indica<tion of the protein cannt be only on the graph at the end of the figure. This is not easy to follow.
  • Please check the unmethylated / demethylating terms
  • Figure 5: Please move the figure at the end of the paragraph. In panel A, please reformat the second graph as the axis cut cut out one of the bar. In panel B, it is not clear how the Unmethylated/Methylated
    index is calculated, it is not mentioned neither in MM, nor in the text, nor in the Figure legend. Please clarify. Please include error bars in the graph in panel B, otherwise is difficult to understand the statistycal analysis.

Discussion: as already said discusion is well done and written. However it would be advisable to better connect the discussion of the different inducers so that it result less as a mere list.

Author Response

The present manuscript is interesting and describe an interesting methodology to determine the immune population present in cancer tissue as well as the potential antitumor role of molecular iodine.

Introduction and discussion are well presented and structured, the subject flow and the background and future directions are well discussed.

However, Material and Methods and Results section must be improved.

Material and Methods:

In the Real Time RT-PCR it is not clear how the authors calculated the relative gene expression. If a comparative DDCt method was used then normalization with b-actin is used; otherwise, if a standard curve was used, then an absolute quantification was done. Please clarify and correct accordingly.

Yes, we use DDCt method using as normalized gene b-actin. We will be mentioned in the new version of Ms.

 A Statistical analysis paragraph should be added

 We thank the referee for the observation, in the new version we include the statistical analysis paragraph

Results:

 Reference to Figure 1 is missing in the text

Figure 1 is difficult to understand. First of all, images quality must be improved, and fonts of the figure must be enlarged as it is impossible to read the gene names. It is not clear what the two columns of each heatmap refer to, a table over the graph should be added.

We thank the referee for the observation. We review and modify the results and figures to make clearer the description.

Figure 1 legend is not self-explanatory and some information are missing. Color code of the heatmap are not clear: does withe means that the gene is not overexpressed and thus not regulated in tumor?

Then how can the authors state that "Genetic overexpression was found to be twofold higher in advanced tumors than in early-stage tumors"? Additionally, as said, it is not clear what refers to early-stage tumor, what to advanced tumor, what to I2 treatment and what to Cht+I2: this is very confusing.

We make changes to the description of the figure 1. We add the group name at the top of each column (I2 and Cht + I2). We also put an explanation of the kay color (intensity of overexpression) and included an explanation of the relationship between I2 and early state and Cht + I2 and advanced state. In the final manuscript, the quality of the figures will be superior, and the names of the genes will be easily readable.

FIGURE 2

Sentence: "I2 increases the intratumoral ratio of antigen-presenting macrophages/dendritic cells...": nowhere a ratio between antigen-presenting macrophages/dendritic cells has been measured or expressed. Please remove the word ratio

Sentence: "With both methods, we observed that the treatments (I2 and Cht) are accompanied by an increase in the levels of M2 macrophages...." It is not correct to say with both methods, as for what I could understand, the ICTD methods cannot discriminate between macrophages subpopulations.

Also figure 2 is difficult to understand and it is not clear what the columns of the two panels refer to: What the two columns for each treatment refer to? Please indicate it more clearly in the graph. Where are the reference to early stage and late stage groups? it is not clear. It was very hard to realte what said in the text with what was in the figure. Please try to facilitate the reading by making the figure easier to understand and the text more explanatory.

We made significant changes to the description of figure 2. We described that the two columns correspond to two individual pools (5 or 6 samples) from each group. We changed the figure for another that contained the names of each group in each column and described the changes with greater precision. We changed the word ratio for relative percentage.

 Figure 4: please indicate which protein is analysed in the first line of IHC and which in the second line of IHC. The indication of the protein cannot be only on the graph at the end of the figure. This is not easy to follow.

Results correspond to early-state samples. We add the full description in description and figure legend.

Please check the unmethylated / demethylating terms

The referee made a good point. We changed the section tittle to clarified the results

Figure 5: Please move the figure at the end of the paragraph. In panel A, please reformat the second graph as the axis cut cut out one of the bar. In panel B, it is not clear how the Unmethylated/Methylated index is calculated, it is not mentioned neither in MM, nor in the text, nor in the Figure legend. Please clarify.

Please include error bars in the graph in panel B, otherwise is difficult to understand the statistical analysis.

Thanks to the referee for the suggestions. We cannot cut the second bar because values less than 1 are lost if the axis is 0-5.

Discussion: as already said discussion is well done and written. However, it would be advisable to better connect the discussion of the different inducers so that it results less as a mere list.

Thanks for your consideration. Taking into account your comments, as well as those of referee 1 who highlight the written discussion, we consider that subdividing it could lose the integrative vision of our results. Therefore, we wish to leave it as it is.